# Genomic epidemiology of *Plasmodium knowlesi* reveals putative genetic drivers of adaptation in Malaysia

Jacob A. F. Westaway [1,2]*, Ernest Diez Benavente[3], Sarah Auburn[1], Michal Kucharski[4], Nicolas Aranciaga[4], Sourav Nayak[4], Timothy William[5], Giri S. Rajahram[5,6], Kim A. Piera[1], Kamil Braima[1], Angelica F. Tan[1], Danshy A. Alaza[5], Bridget E. Barber[1], Chris Drakeley[7], Roberto Amato[8], Edwin Sutanto[9], Hidayat Trimarsanto[1,10], Jenarun Jelip[11], Nicholas M. Anstey[1], Zbynek Bozdech[4], Matthew Field[1,2,12,13], Matthew J. Grigg[1]

**1** Global and Tropical Health, Menzies School of Health Research and Charles Darwin University, Darwin, Northern Territory, Australia, **2** Centre for Tropical Bioinformatics and Molecular Biology, James Cook University, Cairns, Queensland, Australia, **3** Laboratory of Experimental Cardiology, University Medical Center Utrecht, Utrecht, the Netherlands, **4** Nanyang Technological University, Singapore, Singapore, **5** Infectious Disease Society Kota Kinabalu, Kota Kinabalu, Sabah, Malaysia, **6** Queen Elizabeth II Hospital-Clinical Research Centre, Ministry of Health, Kota Kinabalu, Sabah, Malaysia, **7** London School of Hygiene & Tropical Medicine, London, United Kingdom, **8** Wellcome Sanger Institute, Hinxton, United Kingdom, **9** Exeins Health Initiative, Jakarta, Indonesia, **10** Eijkman Research Center for Molecular Biology, National Research and Innovation Agency, Jakarta, Indonesia, **11** Universiti Malaysia Sabah, Kota Kinabalu, Sabah, Malaysia, **12** College of Public Health, Medical and Veterinary Science, James Cook University, Townsville, Queensland, Australia, **13** Immunogenomics Lab, Garvan Institute of Medical Research, Darlinghurst, New South Wales, Australia

☯ These authors contributed equally to this work.
* Jacob.westaway@menzies.edu.au

## Abstract

Sabah, Malaysia, has amongst the highest burden of human *Plasmodium knowlesi* infection in the world, associated with increasing encroachment on the parasite's macaque host habitat. However, the genomic make-up of *P. knowlesi* in Sabah was previously poorly understood. To inform on local patterns of transmission and putative adaptive drivers, we conduct population-level genetic analyses of *P. knowlesi* human infections using 52 new whole genomes from Sabah, Malaysia, in combination with publicly available data. We identify the emergence of distinct geographical subpopulations within the macaque-associated clusters using identity-by-descent-based connectivity analysis. Secondly, we report on introgression events between the clusters, which may be linked to differentiation of the subpopulations, and that overlap genes critical for survival in human and mosquito hosts. Using village-level locations from *P. knowlesi* infections, we also identify associations between several introgressed regions and both intact forest perimeter-area ratio and mosquito vector habitat suitability. Our findings provide further evidence of the complex role of changing ecosystems and sympatric macaque hosts in Malaysia driving distinct genetic changes seen in *P. knowlesi* populations. Future expanded analyses of evolving *P. knowlesi* genetics and environmental drivers of transmission will be important to guide public health surveillance and control strategies.

**Data availability statement:** Genomic data produced as part of this work are available at the Sequence Read archive (SRA) of the National Center for Biotechnology Information (NCBI) under the BioProject ID PRJNA1066389 and all bioinformatic and analytical scripts are available at https://github.com/JacobAFW/Pk_Malaysian_Population_Genetics. Publicly available data used as part of our analyses can be found at https://doi.org/10.1038/s41598-019-46398-z and https://doi.org/10.3201/eid2608.190864

**Funding:** Sample collection and sample processing were supported by the Ministry of Health, Malaysia (grant number BP00500420 and grant number BP00500/117/1002 to GSR); the Australian National Health and Medical Research Council (grant numbers 496600, 1037304 and 1045156 to NMA); the US National Institutes of Health (grant numbers R01AI116472-03 to TW and 1R01AI160457-01 to GSR), and the UK Medical Research Council, Natural Environment Research Council, Economic and Social Research Council, and Biotechnology and Biosciences Research Council (grant number G1100796 to CD). Whole genome sequencing was supported by a Singaporean Ministry of Education Grant (grant number MOE2019-T3-1-007 to ZB), and salary support for bioinformatics and analyses through an Australian NHMRC Ideas Grant (grant number APP1188077 to MG). MF (grant number APP5121190) and MJG (grant number 2017436 ) were supported by NHMRC Emerging Leader 2 fellowships; MJG was also supported by the Australian Centre for International Agricultural Research and Indo-Pacific Centre for Health Security, Department of Foreign Affairs and Trade, Australian Government funded ZOOMAL project (grant number LS/2019/116 to MJG). This research has also been funded by the Australian Government through the Partnerships for a Healthy Region (PHR) initiative (RESPOND project, grant number 79233 to MJG). The views expressed in this publication are the author's alone and are not necessarily the views of the Australian Government. The funders had no role in study design, data collection and analysis, decision to publish, or preparation of the manuscript.

**Competing interests:** The authors have declared that no competing interests exist.

## Author summary

The zoonotic *P. knowlesi* parasite is an emerging, yet understudied, cause of malaria in Southeast Asia. Sabah, Malaysia, has amongst the highest burden of human *P. knowlesi* infection in the world, however, the region is currently understudied. We produced a collection of high-quality *P. knowlesi* genomes from Sabah, and in combination with publicly available data, performed an extensive population genetics analysis. Our work contributes novel insights for *Plasmodium knowlesi* population genetics and genetic epidemiology.

## Introduction

Zoonotic transmission of the macaque parasite *Plasmodium knowlesi* has emerged as the most common cause of human malaria in Malaysia and parts of western Indonesia [1–3]. *P. knowlesi* infections can cause severe, life-threatening malaria, with a case fatality similar to that of *P. falciparum* in Southeast Asia despite comparatively lower levels of parasitemia [4,5]. The recent increased reporting of *P. knowlesi* infections in Southeast Asia has been strongly linked with the encroachment of humans on previously intact habitats of their natural macaque reservoir hosts [6]. Zoonotic transmission of *P. knowlesi* is thought to occur largely in response to increasingly fragmented landscapes as a result of land clearing and associated agricultural activities, with increased exposure in at-risk workers and local populations in endemic areas to both pig-tailed (*Macaca nemestrina)* and long-tailed (*M. fascicularis)* macaques, and the *Anopheles* Leucosphyrus Group mosquito vectors [7,8]. Worryingly, in contrast to the control of other human *Plasmodium* species, national WHO malaria elimination goals in Southeast Asia are threatened by the inability of public health measures to target macaque host reservoirs for *P. knowlesi* [2]. Furthermore, conventional prevention measures such as insecticide-treated bed nets used successfully for other *Plasmodium* species in the region are limited for *P. knowlesi* zoonotic infections, primarily acquired at the forest-edge during agricultural work activities [9,10].

Insights gained from genomic analyses of human malaria parasites have advanced our understanding of basic disease biology, drug resistance and malaria epidemiology [11]. Large-scale, collaborative efforts to produce publicly available population-level whole genome data for *Plasmodium* species of interest, have produced over 20,000 *P. falciparum* [12] and ~1,800 *P. vivax* [13] genomes. In contrast, *P. knowlesi* currently has fewer than 200 whole genomes available from a limited geographic distribution [14–18]. Only 16 reported *P. knowlesi* genomes are described from the state of Sabah in East Malaysia, despite this area representing among the highest reported number of *P. knowlesi* cases and disease burden globally to date [19].

Previous studies of *P. knowlesi* population genetics in Malaysia have identified three genetically divergent populations using a combination of whole-genome sequencing [20] and microsatellite genotyping [21]. One of these populations is restricted to Peninsular Malaysia, whilst the other two are found in Malaysian Borneo. The two overlapping clusters in Malaysian Borneo are derived from the separate macaque reservoir hosts: from long-tailed macaques (*Macaca fascicularis* [*Mf*]) and pig-tailed macaques (*M. nemestrina* [*Mn*]) [22]. We refer to these clusters as *Mf* (cluster 1), *Mn* (cluster 2) and *Peninsular* (cluster 3) throughout this manuscript. Despite these clearly-defined, genetically divergent populations, previous work further identified distinct subpopulations within the different clusters [15], with evidence of recent positive directional selection [20] and large genetic introgression events between the subpopulations linked to mosquito vectors [15]. In this context, introgression

refers to the transfer of genetic information from one cluster to another, resulting from hybridisation and repeated backcrossing. This evidence suggests that *P. knowlesi* population structure is changing, with changes hypothesised to occur as a result of rapidly altering forest and agricultural ecosystems in Malaysia.

To expand our understanding of the evolving population structure of *P. knowlesi* across Malaysia, we performed whole genome sequencing on 94 new human infections from diverse landscapes across Sabah, East Malaysia [19]. The newly produced data were combined with 108 (100 included in the analysis) publicly available *P. knowlesi* genomes derived from clinical infections across Malaysia [14,20]. Leveraging the additional isolates from Sabah, our objective was to perform a comprehensive evaluation of *P. knowlesi* population structure with a dataset that better represents the distribution of symptomatic infections from passive case detection across Malaysia. We combined genomic data with environmental land cover classification data surrounding knowlesi malaria case villages to better explore the relationship between the genomic and ecological features in Sabah associated with the transmission of *P. knowlesi* populations. These integrated analyses aim to provide insights to assist in the development of future public health interventions and genomic surveillance efforts.

## Methods

### Ethics statement

The research was performed in accordance with the Declaration of Helsinki and ethics approval was obtained from the medical research ethics committees of the Ministry of Health, Malaysia and Menzies School of Health Research, Australia.

### Sample collection and preparation

We used a combination of newly generated *P. knowlesi* whole genome sequencing data (n = 94) [23] and archived FASTQ files (n = 108) from *P. knowlesi*-infected patients in Malaysia. Newly processed samples were collected as part of prospective clinical studies conducted through the Infectious Diseases Society Kota Kinabalu Sabah-Menzies School of Health Research collaboration from 2011 to 2016 across multiple hospital sites in Sabah [4,5]. Patients of all ages presenting with microscopy-diagnosed malaria were enrolled following informed consent. Single species, *P. knowlesi* infections were confirmed through validated PCR (targeting the 18S small-subunit RNA gene) [24,25] and parasitemia quantified by expert research microscopists. These 94 clinical isolates underwent Illumina whole genome, paired-end sequencing (150 bp), with library preparation conducted using the NEBNext Ultra IIDNA Library Prep Kit (from New England BioLabs Inc., Cat No. E7645). The further 108 samples from a broader geographic range within Malaysia were downloaded from the National Center for Biotechnology Information (PRJEB33025, PRJEB23813, PRJEB1405, PRJEB10288 & PRJN294104) [14,20] (Table A in S1 Text).

### Read mapping, variant discovery and genotyping

Variants were detected using a modified version of a previously described workflow [26]. Raw reads were processed using FastQC and cutadapt [27] to determine quality, with subsequent filtering and trimming of reads. The Burrows-Wheeler Aligner (bwa) was then used to map reads to the PKA1-H.1 reference genome [28]. BAM pre-processing steps were applied using Picard version 2.26.1 and the Genome Analysis Toolkit (GATK) version 3.8-1-0 [29]. Notably, two steps in the GATK workflow (base recalibration and indel realignment) require a set of high-quality known variants. As recommended by the Broad Institute for non-model organisms without a reference dataset [30], we took a bootstrapping approach, where we passed a

subset of 39 samples through a simplified version of the pipeline and applied hard filtering based on quality score distribution (FS<=2, MQ>=59 & QD>=20), and then passed this conservative variant set into the recalibration steps of the pipeline for another round of variant calling.

SNPs and indels were called using a consensus approach applied to outputs from GATK and bcftools variant callers using a modified version of a previously described workflow [31,32]. A consensus approach was taken to improve accuracy and reduce false positives by filtering for the overlap between the two commonly used tools, which use distinct algorithms and heuristics. For GATK, HaplotypeCaller was used to identify potential variants in each sample, with the resulting GVCF files merged using CombineGVCFs, and joint-genotyping performed using GATK's GenotypeGVCFs. A similar joint-calling approach was implemented with bcftools using the mpileup and call subcommands. A consensus was taken of the resulting VCF files generating a conservative list of high-quality variants. Finally, SNPs and small indels were filtered using GATK's VariantFiltration using the same thresholds outlined above.

### Data filtering

To reduce noise, errors, and avoid bias in statistical estimates from rare variants, further filtering was applied based on clonality ($F_{ws}$), genotypic missingness and minor allele frequency (MAF), depending on the downstream analysis. For clonality, within-isolate fixation index $F_{ws}$ [33] was calculated on the full dataset (n = 201) using the moimix package (github.com/bahlo-lab/moimix) and samples with $F_{ws}<0.85$ were removed from downstream analyses [15]. $F_{ws}$ is a measure of genetic diversity within an isolate, where the genetic variation within individuals is compared to the genetic variation across and entire population. Prior to filtering samples based on $F_{ws}$, the non-reference allele frequency (NRAF) was also plotted across the genome for individual samples using ggplot2 [34]. Genotypic missingness and MAF were then calculated using PLINK2 [35,36]. SNPs with MAF <5% or genotypic missingness >25% across the population, and samples with >25% genotypic missingness, were filtered from downstream analyses, as well as those SNPs located in hypervariable regions (Table B in S1 Text).

### Characterising population structure

To determine overarching population structure, several complementary strategies were employed including neighbour joining analysis based on identity-by-state (IBS), connectivity based on identity-by-descent (IBD) [37], and ADMIXTURE analysis [38]. IBS, a measure of genetic similarity where two alleles at a given locus are identical, was calculated with PLINK and visualised with neighbour-joining trees (NJT) [39] in R using ggplot2 and ggtree [40]. IBD, a measure of genetic similarity where alleles are considered identical if they were inherited from a common ancestor, was calculated with hmmIBD [41], which implements a hidden Markov model to determine sequence segments of shared ancestry. Base R and igraph [42] were used for IBD visualisation at a variety of thresholds (represent the percentage of the genome that is IBD between pairs of samples). To determine the proportions of mixed ancestry, ADMIXTURE was used to implement a maximum likelihood estimation, which was then visualised in R using ggplot2. CV error was calculated prior to ADMIXTURE analysis to identify the optimal K value. As K=3 was deemed optimal, exhibiting a low cross-validation error compared to other K values determined by ADMIXTURE's cross-validation procedure, and the distribution of samples aligned with the NJT, the K clusters were referred to throughout the manuscript with the previously defined Peninsular- and macaque-associated cluster names (*Macaca fascicularis* (*Mf*), *Macaca nemestrina* (*Mn*) & *Peninsular*).

## Identifying the presence of introgression events

We performed a bespoke analysis to identify possible genomic regions of introgression. First, we identified the major allele for each cluster at each genomic coordinate. Then using a sliding window approach (10kb windows), we determined the genetic distance for each sample to each cluster. Genetic distance was defined as the proportion of mismatched SNPs per sliding window (10kb) when comparing the called allele in the sample to the major allele for a cluster at each position. The genetic distances were then plotted on a two-dimensional axis, with different clusters along the x and y axis, and two-dimensional kernel density estimations (contours) were calculated for the genomic clusters using ggplot2 (github.com/tidyverse/ggplot2) and MASS (github.com/cran/MASS) packages, and the density contours overlayed on the plot. The *points.in.polygon* function (github.com/edzer/sp) was used to determine in which contours the windows are spatially located. Windows located within the contours of another cluster whilst outside the major contours of their own were defined as introgressed. To be conservative, candidate windows underwent several filtering steps, including those that appear in>= 5% of the population and the removal of windows that overlap hypervariable regions (Table B in S1 Text).

## Exploring links between introgression and environmental land types

We performed subsequent regression analyses to explore whether surrounding village-level environmental land types and predicted vector habitat suitability are associated with *P. knowlesi* introgression. The primary residential addresses for deidentified *P. knowlesi* cases for the preceding 3 weeks before health facility presentation was first used to obtain centroid village-level location coordinates cross-checked for accuracy using Google Earth (version 7.3). Selected environmental classification metrics of forest fragmentation (percentage of Landscape – tree cover and Perimeter-Area Ratio – tree cover) within a 5km radius surrounding village locations were then calculated from a composite landscape metrics tool encompassing ESRI 2020 and Sentinel-2 GIS data at 10-metre resolution [43] (Figs A and B in S1 Text). The relative predicted *Anopheles* Leucosphyrus Complex mosquito vector occurrence surface from Moyes et al. [44] based on boosted regression tree models encompassing mosquito sampling presence/absence data (1999-2014) and environmental covariates indicating habitat suitability was obtained through the *malariaAtlas* R package [45]. The mosquito vector habitat suitability surface was averaged within a 5x5km grid around the geolocated village sites. Moran's I [46] was calculated to exclude spatial autocorrelation of the environmental land types and predicted vector habitat suitability at the selected grids. Univariate regression analyses were initially used to assess potential associations between these environmental parameters and the macaque-derived clusters. Tertiles were then generated representing the degree of introgression in samples, with samples categorised as having *low*, *medium*, or *high* introgression. Logistic regression models were subsequently implemented to assess if landscape fragmentation indices or *Anopheles* Leucosphyrus Complex habit suitability were associated with either the presence of the top ten most frequently introgressed windows (binomial) or the degree of introgression (ordinal). The Akaike Information Criterion (AIC) [47] was compared to determine the optimal model design (scripts available in the attached GitHub repository).

## Identification of orthologous antimalarial drug resistance markers

Antimalarial drug resistance markers for *P. falciparum* and *P. vivax* (putative) were collated using multiple sources [48,49] and their orthologues in *P. knowlesi* identified. This includes dihydrofolate reductase (*dhfr*), dihydropteroate synthase (*dhps*), chloroquine resistance transporter (*crt*), multidrug resistance protein 1 (*mdr1*), multidrug resistance-associated proteins 1 (*mrp1*), plasmepsin 4 (*pm4*), kelch 13 (*k13*), reticulocyte binding protein 1a (*rbp1a*) and

reticulocyte binding protein 1b (*rbp1b*). *P. falciparum* and *P. vivax* orthologues were identified using PlasmoDB's [50] *Orthologue and synteny* tool, which is based on the OrthoMCL database, a genome-scale algorithm for grouping orthologous protein sequences [51]. Multiple sequence alignment was then performed on PlasmoDB sequences, comparing the *P. knowlesi* amino acid sequences against the relevant orthologues in *P. falciparum* and *P. vivax* to identify shared mutations between orthologue genes.

## Characterising subpopulations within Malaysian Borneo

Once the larger *P. knowlesi* population structure was characterised, we investigated differences between sub-population clusters within Malaysian Borneo and interrogated the genomes of all relevant samples for signs of differentiation. Samples were subset to the major genomic clusters (*Mf* and *Mn*) of Malaysian Borneo, using their distribution on the NJT and major population assigned by ADMIXTURE. Then, both IBS and IBD analyses were repeated on these subsets (*Mf* and *Mn* specific subsets) to determine whether subpopulations exist within these larger populations. PLINK was used to calculate the fixation index ($F_{ST}$), a measure of population differentiation due to genetic structure, specifically, the variance of allele frequencies between populations. ggplot2 was used to visualise $F_{ST}$ across the genome using a non-overlapping sliding window (1-kb) approach. This allowed the identification of outlier regions, which were annotated to identify potential genes of interest. Further details and scripts for the methods described can be found at https://github.com/JacobAFW/Pk_Malaysian_Population_Genetics.

## Results

The 94 newly sequenced *P. knowlesi* whole genomes all originated from the state of Sabah, encompassing human infections from 11 administrative districts, including 22 infections from Kota Marudu and 14 from Kudat (Table A in S1 Text), collected between May 2011 and February 2016. These genomes had an average of 66,415,402.53 reads per sample, with 30.27% mapping to the PKA1-H.1 reference genome [28]. The average sequencing depth (excluding mitochondria and apicoplast) of these new genomes was 80X (2 to 362X across samples), with 82.5% (18.9 to 97.2% across samples) of the bases in the reference genome covered. The distribution of reads across chromosomes was relatively even, with the mean sequencing depth ranging from 70 to 87X, and the mean percentage of bases covered ranging from 80.3% to 84.0%. The 108 high-quality publicly available *P. knowlesi* genomes from NCBI were derived predominantly from human infections (with six laboratory strains passaged through macaques) across different districts from both Peninsular Malaysia (n=33) and East Malaysia on the island of Borneo, including the geographically distinct neighbouring state of Sarawak (n=59) in addition to a small number from Sabah (n=16). $F_{WS}$ analysis was performed on 201 samples, with additional filtering based on clonality, missingness, and minor allele frequency, used to generate a subset for other analyses. 52 of the newly sequenced genomes remained after the additional filtering, and another 100 genomes from the publicly available data. The combined 152 *P. knowlesi* genomes consisted of: 61 from Sabah, 59 from Sarawak, and 32 from Peninsular Malaysia. Joint genotyping initially identified 1,542,627 single nucleotide polymorphisms (SNPs), which after filtering (clonality, missingness and minor allele) resulted in 357,379 SNPs.

### Complex *P. knowlesi* infections in Malaysian Borneo

Given that *P. knowlesi* parasites are haploid in the blood stage of host infection, the presence of multiple alleles at given loci is indicative of a multiple clone (polyclonal) infection. The within-isolate fixation index ($F_{WS}$) was used to measure the genetic complexity of all infections (n = 201).

The $F_{WS}$ score ranges from 0 to 1, with increasing values reflecting increasing clonality [52]. At a commonly applied threshold of $F_{WS} < 0.95$, 13.4% (n=27/201) of infections were polyclonal. The highest proportion of polyclonal infections was observed in Sarawak (17.6%, n=13/74), followed by Peninsular Malaysia (12.1%, n=4/33) and Sabah (10.6%, n=10/94). Sarawak had significantly lower $F_{WS}$ than both Sabah (p < 0.05) and Peninsular Malaysia (p < 0.05) (Figs 1A and C in S1 Text).

Non-reference allele frequency (NRAF) plots illustrate a variety of within-host diversity patterns. This includes distinct clones (18.5% [n=5/27]) (e.g., ERR985376 [$F_{WS}$ = 0.93]) and genetically mixed clones (e.g., ERR985395 [$F_{ws}$ = 0.54]). Samples with distinct clones could be the result of either superinfection (multiple mosquito inoculations) or co-transmission (single mosquito inoculation). However, those that are genetically mixed are likely the result of co-transmission events, as given adequate genetic diversity in a population (not inbred), superinfections with highly related clones are unlikely.

## *P. knowlesi* genomes from Sabah belong predominantly within the *Mf* cluster

Previous genetic studies have described distinct genetic clustering of *P. knowlesi* into a geographic Peninsular-Malaysia sub-population, and two Malaysian-Borneo macaque-associated subpopulations; *M. fascicularis* (*Mf*) and *M. nemestrina* (*Mn*) [17,22,21]. We sought to determine the genetic clustering patterns of the Sabah genomes relative to infections from Sarawak and Peninsular Malaysia. Neighbour-joining analysis based on identity-by-state (IBS) was undertaken on the 152 low complexity *P. knowlesi* genomes from across Malaysia, revealing

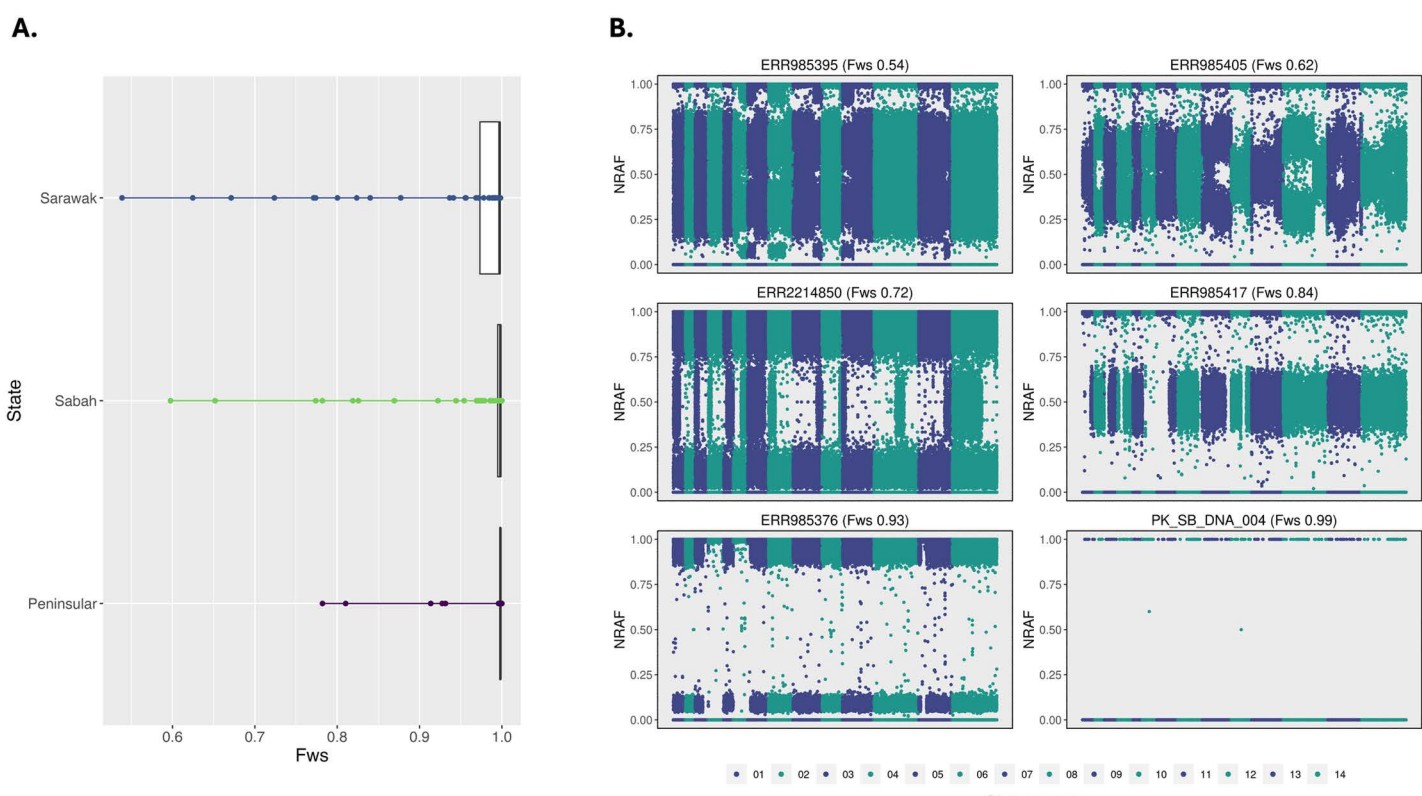

**Fig 1. Comparable within-isolate genetic complexity across geographic regions.** A: Boxplots depicting the distribution of within-infection diversity ($F_{WS}$) across three regions (*Sabah*, *Sarawak*, and *Peninsular*). B: Dot plots of the within-isolate non-reference allele frequencies (NRAF) across the genome for six Malaysian *P. knowlesi* infections ranging from low diversity ($F_{WS}$=0.99) to high diversity ($F_{WS}$=0.54) and with varying levels of within-host relatedness.

three clusters (Fig 2A). The newly sequenced *P. knowlesi* samples originating from Sabah group predominantly within the *Mf* cluster (82.7%, n=43/52); the remaining (17.3%, n=9/52) infections clustered within the *Mn* clade, similar to the proportions of the 164 samples from Sabah previously described by Divis et al. 2017 [21] (*Mf* = 86.6%, *Mn* = 13.4%) [21]. ADMIXTURE analysis revealed the greatest likelihood of 3 sub-populations amongst the 152 infections (Fig F in S1 Text), confirming the patterns observed with neighbour-joining analysis (Fig 2C).

Despite substantial genetic divergence between the *Mf* and *Mn* clusters (mean $F_{ST}$ = 0.2), there is also substantial geographic overlap (Fig 2B) and evidence of shared ancestry (>1% ancestry to two or more groups) amongst 6.6% (n=10/152) of infections (Fig 2C). This observation extends beyond the newly sequenced Sabah samples and to those previously reported in neighbouring Sarawak, with the separate genomic *Mf* and *Mn* clusters and several samples of *Mf* and *Mn* ancestry being identified in both geographic locations. *P. knowlesi* infections with shared ancestry originated from the Sarikei and Betong districts in the state of Sarawak and four of the newly sequenced Sabah infections (from Papar, Ranau and Kota Marudu districts). Although several Malaysian-Borneo *P. knowlesi* infections had evidence of shared ancestry, samples from Peninsular-Malaysia and the district of Kapit in Sarawak (both *Mn* and *Mf*) are descendants of single ancestral populations (Fig 2C).

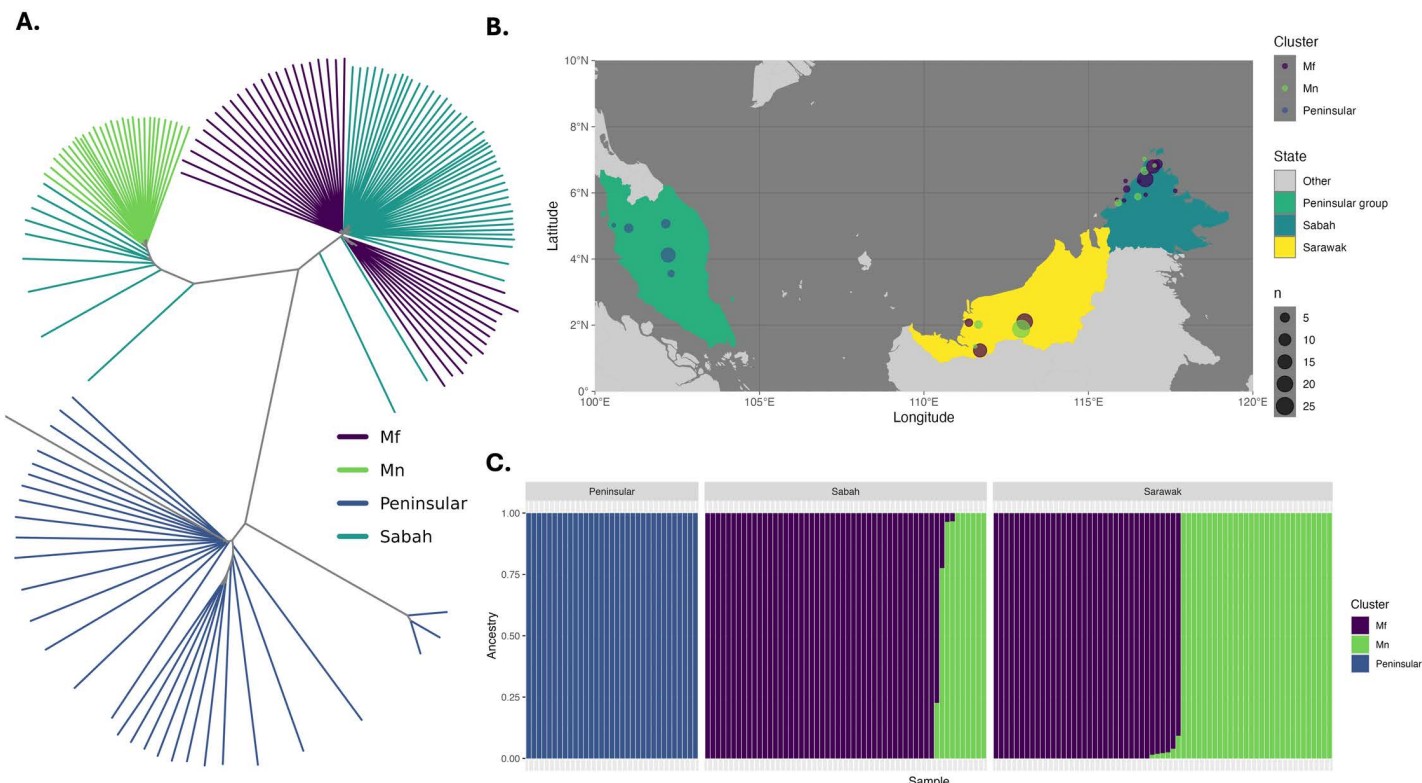

**Fig 2. Geographic overlap and evidence of shared ancestry between the *Mf* and *Mn* P. knowlesi clusters.** A: Unrooted neighbour-joining tree based on identity-by-state (IBS) depicting three predominant genomic clusters of *P. knowlesi* across Malaysia, specifically the Peninsular Malaysia sub-population (*Peninsular*), and Malaysian-Borneo macaque-associated subpopulations of *Macaca fascicularis* (*Mf*) and M. nemestrina (*Mn*). The new isolates from Sabah are labelled separately (*Sabah*). B: Map of Malaysia showing the geographic distribution and number of samples, and genomic clusters across Malaysian-Borneo (right) and Peninsular-Malaysia (left). C: Bar plot illustrating the proportionate ancestry to each of 3 (K) subpopulations determined by ADMIXTURE for each sample (bars on x-axis), sectioned by geographic region. The three K populations identified aligned perfectly with the clustering in the NJ tree; K=1 with *Mf*, K=2 with *Mn* and K=3 with Peninsular as per the colour-coding. Shapefile made with Natural Earth: https://www.naturalearthdata.com/downloads/50m-cultural-vectors/50m-admin-1-states-provinces/.

## Greater genetic diversity within the *Mf* than *Mn* and *Peninsular* clusters

Since malaria parasites are recombining organisms, neighbour-joining analysis can miss recent connectivity between infections where outcrossing has taken place. To further elucidate the relatedness between isolates, both within and across clusters, we performed identity-by-descent (IBD) analysis on the 152 low complexity infections. In IBD analysis, genomic segments are characterised as identical by descent in pairwise comparisons when identical nucleotide sequences have been inherited from a common ancestor. The *Mf* cluster had the highest genetic diversity, with a median IBD of 7.0%, and as such, we see most of the connectivity break down at a relatively low threshold of 10% (Fig 3). In contrast, the *Mn* (median IBD = 0.5) and *Peninsular* (median IBD = 0.3) clusters maintain tight networks at 25% IBD, reflective of more recent common ancestry and a greater number of shared haplotypes, and in turn, lower transmission intensity (Fig 3). An *Mf* isolate (PK_SB_DNA_028 – Papar, Sabah) also maintains connectivity with the *Mn* cluster at an IBD threshold of 10%, with the regions of IBD between the *Mf* isolate and the *Mn* isolates consistent across pairwise comparisons (Fig G in S1 Text). The *Mf* isolate was collected in Papar (Sabah), and the *Mn* isolates from Kapit, Sarikei and Betong (Sarawak). To confirm the high IBD values in *Mn* and *Peninsular* clusters were not inflated by the SNPs used (i.e., being biased by strong population structure), we trialled multiple filtering combinations and re-calculated the median IBD values for comparison, confirming our initial findings (Table C in S1 Text).

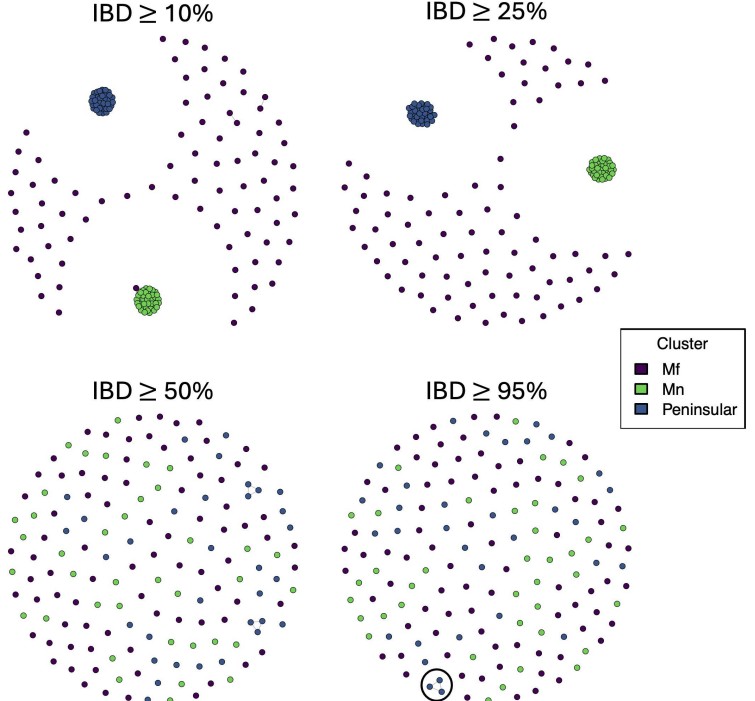

**Fig 3. Identity-by-descent (IBD)-based analysis reveals greater relatedness amongst *Mn* and *Peninsular* than *Mf* samples.** Each circle reflects an infection, colour-coded by genomic clustering group, and the number of lines between infections reflects relatedness (more lines reflect greater relatedness) at the given connectivity threshold of minimum IBD. Where two circles are not connected by a line, the estimated IBD between those infections was below the given threshold. The three samples from Peninsular Malaysia with >95% IBD represent laboratory-based strains from the 1960s that have been passaged through macaques (SRR2222335, SRR2225467 & SRR3135172).

## Greater relatedness within state-level *P. knowlesi* subpopulations

It was hypothesised that *P. knowlesi* clinical infections derived from the separate states of Sabah and Sarawak in Malaysian Borneo are likely to have distinct genetic ancestry due to factors such as differences in the primary *Anopheles* Leucosphyrus Group mosquito vector species and other large-scale environmental features that may have restricted historical gene flow [53]. To test this, we leveraged the newly sequenced genomes to perform additional analyses on a subset of the data comprising isolates from Sabah and Sarawak. We examined the potential impact of geographical regions on population structure and genetic relatedness within each of the separate *Mf* and *Mn* clusters, performing IBD analyses on the clusters separately. IBD analyses of *Mf* and *Mn* subsets suggest that most samples have greater connectivity within their respective states (Fig 4). Two Sabah samples within the *Mf* cluster had a high degree of connectivity with Sarawak samples (Fig 4A). The samples (PK_SB_DNA_028 and PK_SB_DNA_053) are *P. knowlesi* infections collected from residents of the Papar and Kudat districts in Sabah.

## Population differentiation within *Mf* and *Mn* clusters of geographic subpopulations

Given that sampling sites of the *P. knowlesi* geographic subpopulations are isolated by several hundred kilometres, with spatially heterogenous environmental pressures, we performed genome-wide scans for differentiation between the geographic subpopulations within *Mf* and *Mn* subsets. Genome-wide scans within *Mf* highlighted several regions of significant

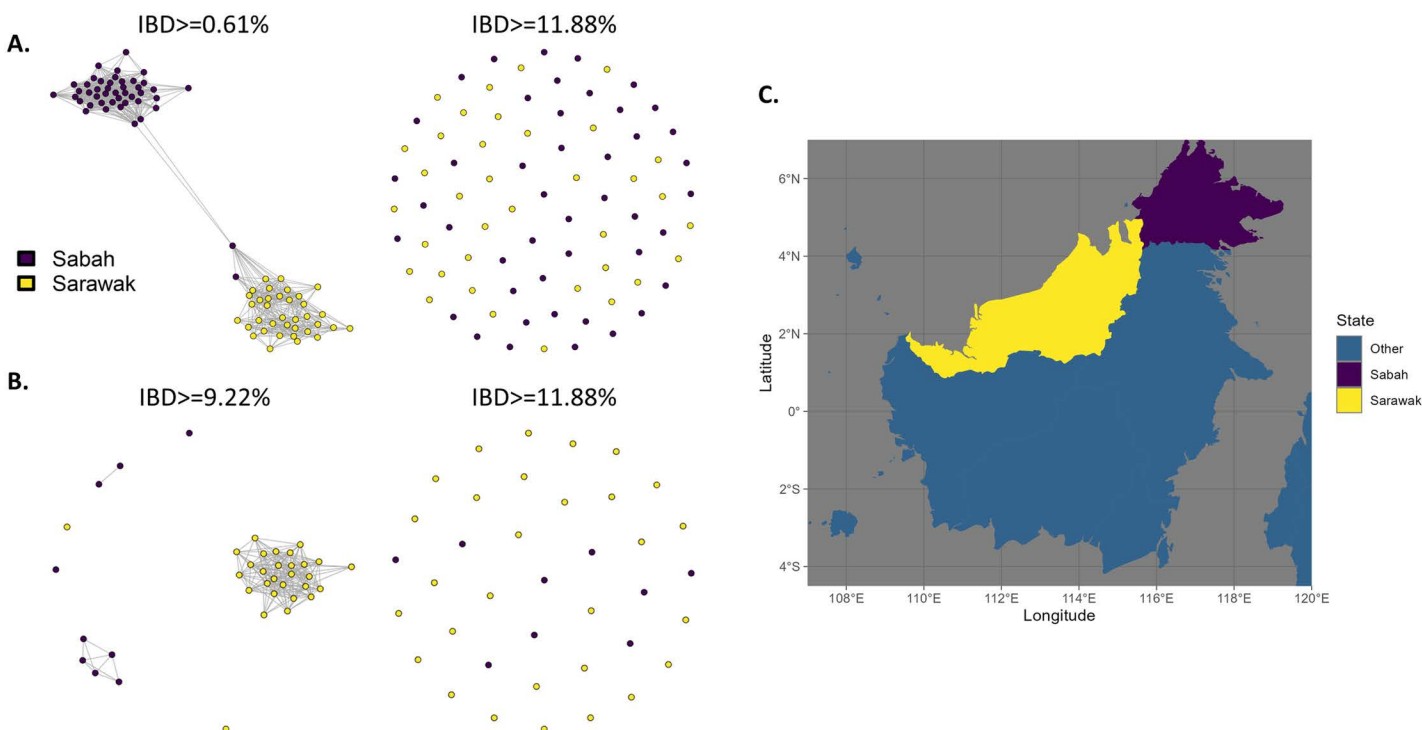

**Fig 4. Infection connectivity is partly driven by geography at the state-level administrative boundary.** A & B: Identity-by-descent (IBD)-based cluster network illustrating the distant relatedness for samples within *Mf* (A) and *Mn* (B) clusters collected in two adjacent states; Sabah and Sarawak, at different cut-offs for the proportion of IBD in a paired comparison. C: Map of East Malaysia on the island of Borneo with colours representing the two states being compared in the IBD analysis. Shapefile made with Natural Earth: https://www.naturalearthdata.com/downloads/50m-cultural-vectors/50m-admin-1-states-provinces/.

differentiation across the genome, appearing as peaks of multiple tightly clustered windows of high $F_{ST}$ against a background of low differentiation (mean $F_{ST}$ = 0.007, Fig 5A). The most notable peaks within the *Mf* cluster were observed on chromosomes 8, 11 and 12. The peak on chromosome 8 covers a region containing the gene encoding for the oocyte capsule protein, with a complete list of genes found in the peak available in Tables H and I in S1 Text. Unfortunately, due to substantial noise, it was not possible to appropriately identify peaks for the *Mn* cluster (mean $F_{ST}$ = 0.036, Fig 5B).

## Substantial evidence for introgression between *Mn* and *Mf* clusters

Previous studies have described the occurrence of chromosomal-segment exchanges between the *Mn* and *Mf* subpopulations, suggesting that they are not genetically isolated [15]. We therefore sought evidence for introgression events in our large collective cohort, and specifically, in the previously underrepresented state of Sabah. Comparisons of genetic distance between 10kb sliding windows in individual *P. knowlesi*-infected samples and different clusters reveal evidence of substantial genetic exchanges across genomic clusters, chromosomes, and geographical regions (Fig 6 and Table D in S1 Text). The degree of introgression, represented by the number of introgressed windows identified in a sample, also varied between all of the above-mentioned features.

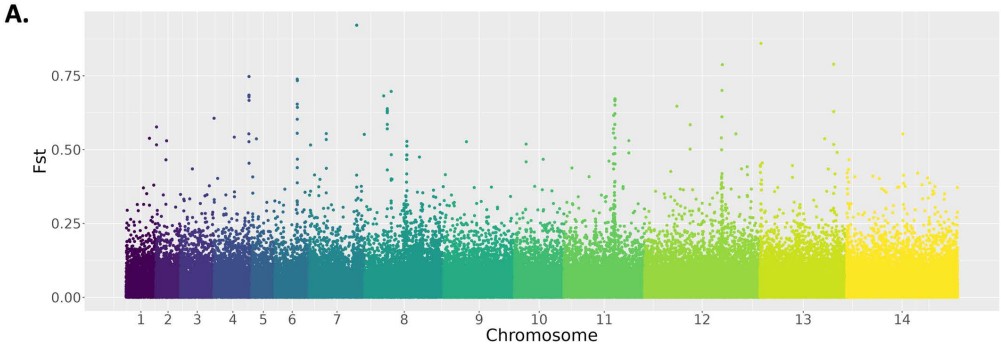

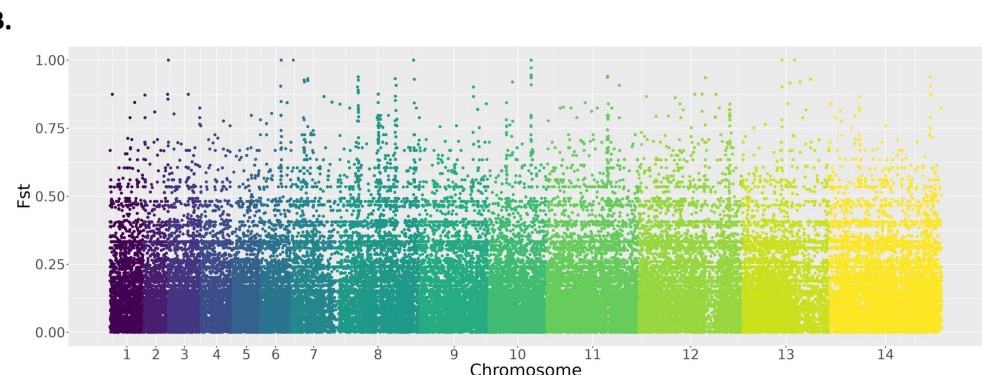

**Fig 5. Genetic differentiation reveals candidate adaptations.** Genome-wide scans of differentiation between Sabah and Sarawak subpopulations within the (A) *Mf* and (B) *Mn* clusters using the between-population fixation index ($F_{ST}$). Only the *Mf* cluster shows clear differences in diversity with peaks of differentiation clear at several chromosomes (most notably on chromosomes 8, 11 and 12), whilst the *Mn* cluster has substantial 'noise' across the genome, with high levels of differentiation across the genome.

Of the 152 individual *P. knowlesi* samples analysed, 71.1% (111/152) had introgressed windows (10kb). Given the complexity of distinguishing single biological introgression events, due to potential ambiguity when grouping adjacent windows with varying start and stop positions across samples (Fig H in S1 Text), we define the 10kb windows identified in our analysis as introgressed windows without making assumptions about the underlying biological events. Approximately 29.5% (n=46/152) of samples demonstrated a high degree (>5 10kb windows) of introgression. Within the subset of newly generated genomes from Sabah, 82.7% (n=43/52) samples had two or more introgressed windows, including 20 with >5 windows. The *Mf* cluster had a higher median number of introgressed windows per sample (median 5, IQR $\pm$ 9.27 ) compared to *Mn* (median 1, IQR $-0.88$ ). Introgressed windows across ten chromosomes for *Mf* and six chromosomes for *Mn*. For the *Mf* cluster, chromosomes 8 (n = 35) and 11 (n = 21) had the greatest number of introgressed windows, with several windows on chromosome 8 overlapping the large peak observed in the $F_{st}$ analysis (Fig 5). For *Mn*, all six chromosomes contained a single window.

The district of Betong in Sarawak had the highest median number of introgressed windows per individual *P. knowlesi* sample (median 29, IQR $\pm$ 2.2 ) followed by the district of Papar in Sabah (median 19, IQR $\pm$ 25.46 ). 85% (n =12/14) of samples from Betong and 50% (n=1/2) from Papar had high levels of introgression, with all but one Betong sample from the *Mf* cluster. The *Mf* isolate from Papar with high levels of introgression (PK_SB_DNA_028), had the greatest number of windows overall (n = 37), followed by ten *Mf* samples from Betong that had greater than 20 introgressed windows (Table D in S1 Text). This same sample, (PK_SB_DNA_028) is also the *Mf* isolate that shared a higher degree of IBD (10%) with the *Mn* cluster relative to its own cluster (Fig 3), suggesting that introgression events may be a contributing factor to shared regions of IBD between samples (Fig G in S1 Text). Furthermore, this mechanism also explains the IBD-based connectivity between the two Sabah samples (PK_SB_DNA_028 and PK_SB_DNA_053) and Sarawak samples in the cluster-specific IBD analysis (Fig 4A), as both samples also exhibit substantial patterns of introgression, in a similar pattern (the same or proximal windows) to that seen in the samples from Sarawak (Table G in S1 Text).

Several candidate introgressed regions identified in the *Mf* clusters overlap putative genes involved in host interactions. Within Sabah, the most common candidate window (window 1504, chromosome 11: 2080000 - 2089999), observed in both Sabah (n = 17) and Sarawak (n = 7) isolates, overlaps a gene encoding for the parasite DNA repair protein RAD50 (PKA1H_110050600), which may aid survival within the host [54]. Focusing on the top 10 most abundant windows in both Sabah and Sarawak, several other genes encoding for proteins essential for survival or invasion in the human, macaque or mosquito hosts were also identified to overlap candidate windows (Table 1). Amongst windows more prevalent in samples from Sarawak were several genes that encode for mosquito-related proteins. This includes the oocyst capsule protein (PKA1H_080026000), CPW-WPC family protein (PKA1H_080026200) and the microneme-associated antigen (PKA1H_080031400). The inclusion of the new Sabah isolates expands the distribution of the introgression event associated with the oocyst-expressed *cap380* gene, previously only observed in Betong, Sarawak [15,18]. The oocyst capsule protein is essential for the maturation of ookinete into oocyst in *P. berghei* and is assumed to assist in immune evasion in mosquito hosts [55]. The CPW-WPC proteins are zygote/ookinete stage-specific surface proteins and appear to be involved in mosquito-stage parasite development [56] and micronemes are critical for host-erythrocyte invasion [57]. The number of candidate regions in *Mn* was minimal (n = 6), with even fewer involved in host interactions. The exception being window 807 on chromosome 08 (760000-769999), overlapping the PKA1H_080021900 gene, which is essential for erythrocyte invasion

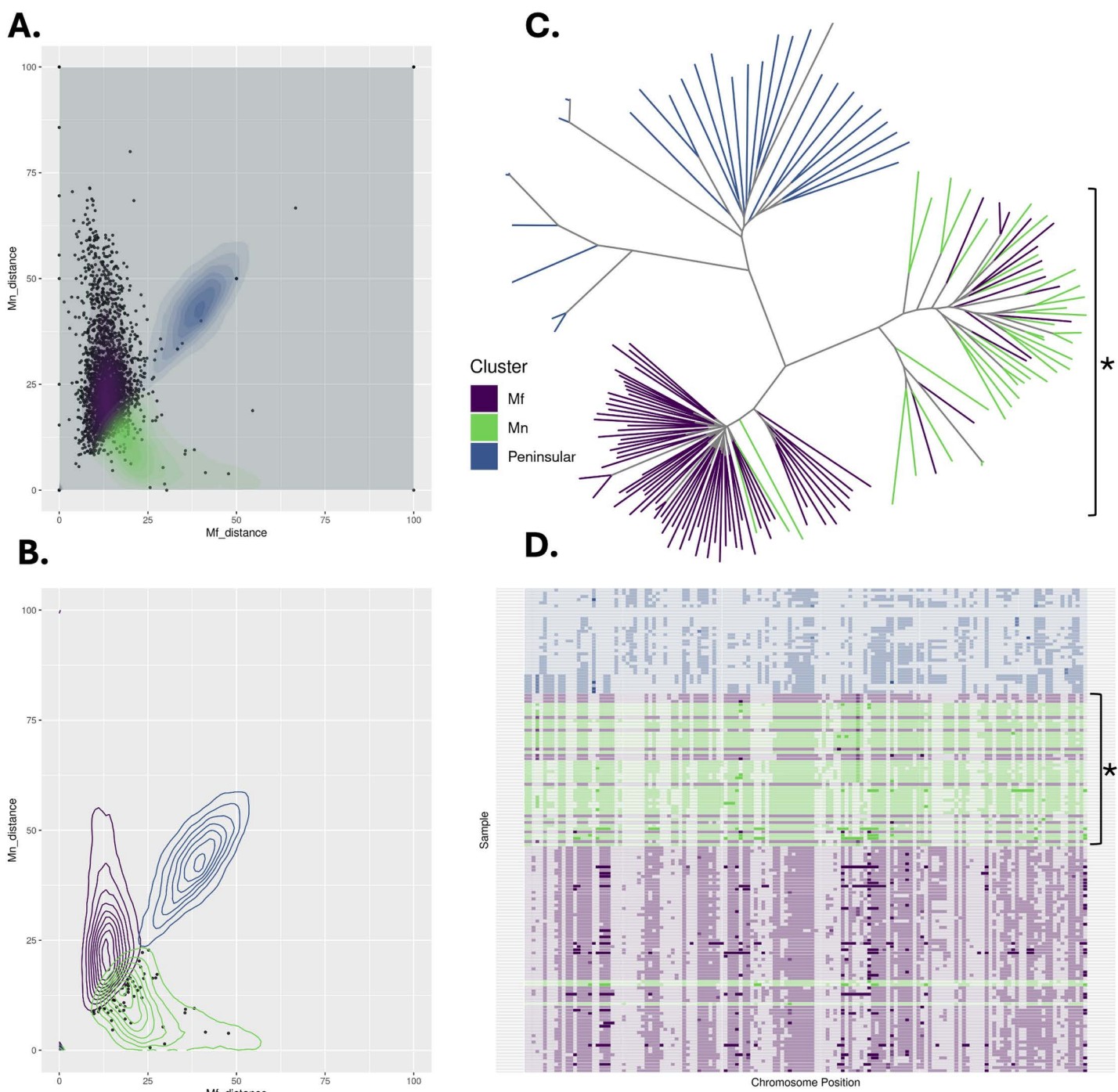

**Fig 6. Determination of introgressed windows between the *Mf* and *Mn* P. knowlesi clusters.** A: Dot and contours plot describing potential introgression events within an *Mf* sample, where x and y axes represent the genetic distance of the sample to the *Mf* and *Mn* clusters, respectively. Genetic distance is the proportion of mismatched SNPs per sliding window (10kb) when comparing the called allele in the sample to the major allele for a cluster at each position. The contours represent the density of genetic distances for the three clusters. B: Dot and contours plot of the same sample above, subset to those windows deemed to be introgressed from the *Mn* cluster. Possible introgression events are sliding windows that fall outside the major contours of the samples own cluster and within the major contours of another, representing greater similarity in genetic distance to the other cluster. C: Unrooted neighbour-joining tree based on identity-by-state (IBS) of *window 1504* on chromosome 08 (950000-959999) and overlapping the PKA1H_080026000 gene (encodes the oocyst capsule protein). The *Mf* samples/branches clustering within the *Mn* branches (depicted by asterisk) provides further evidence that introgression of this window has occurred in these samples. D: SNP barcode plot of *window 1504* on chromosome 08 (950000-959999) showing greater genetic similarity between several *Mf* samples (depicted by asterisk) and the *Mn* cluster, where the colours reflect those in the legend on panel C, and the alpha represents the allele.

**Table 1. Putative genes overlapping the ten most common introgression windows in *Mf* and *Mn*.**

| Window | Gene | Chromosome: start - end | Protein | Sabah (n [%]) | Sarawak (n [%]) |
|---|---|---|---|---|---|
| 1504 | PKA1H_110050600 | 11: 2080000 – 2089999 | DNA repair protein RAD50 | 17 [27.78] | 7 [10.78] |
| 1216 | PKA1H_100020600 | 10: 723,246 – 723,968 | Apicortin | 8 [14.81] | 1 [1.54] |
| 1216 | PKA1H_100020800 | 10: 720000 – 729999 | F-actin-capping protein subunit beta | 8 [14.81] | 1 [1.54] |
| 1223 | PKA1H_100022400 | 10: 790000 – 799999 | NLI interacting factor-like phosphatase | 8 [14.81] | 1 [1.54] |
| 1236 | PKA1H_100025500 | 10: 923,327 – 924,091 | Orotate phosphoribosyltransferase | 8 [14.81] | 2 [3.08] |
| 1359 | PKA1H_110018600 | 11: 629,218 – 635,353 | Patatin-like phospholipase | 8 [14.81] | 7 [10.77] |
| 1309 | PKA1H_110008500 | 11: 138,605 – 141,595 | Secreted ookinete protein | 6 [11.11] | 9 [13.85] |
| 1309 | PKA1H_110008100 | 11: 128,971 – 130,944 | ABC transporter E family member 1 | 6 [11.11] | 9 [13.85] |
| 31 | PKA1H_010010000 | 01: 299,231 – 301,798 | DNA mismatch repair protein MSH2 | 6 [11.11] | 3 [4.62] |
| 826 | PKA1H_080026000 | 08: 940,469 – 950,653 | Oocyst capsule protein | 1 [1.85] | 10 [15.38] |
| 826 | PKA1H_080026200 | 08: 954,493 – 956,175 | CPW-WPC family protein | 1 [1.85] | 10 [15.38] |
| 826 | PKA1H_080026300 | 08: 956,849 – 958,271 | Protein transport protein SEC22 | 1 [1.85] | 10 [15.38] |
| 827 | PKA1H_080026700 | 08: 968,445 – 976,496 | ABC transporter I family member 1 | 1 [1.85] | 10 [15.38] |
| 854 | PKA1H_080031400 | 08: 1,229,556 – 1,230,530 | Microneme associated antigen | 3 [5.56] | 10 [15.38] |
| 819 | PKA1H_080024200 | 08: 878,480 – 880,896 | ATP-dependent RNA helicase DDX6 | 1 [1.85] | 9 [13.85] |

Description of gene name, position, encoded protein, and the isolate counts and proportions for the two Malaysian Borneo states.

in *P. falciparum* [58]. It should be noted that although the biology of these putative genes is well understood in other human-only *Plasmodium* species, they may not directly translate to the biologically and genetically distinct *P. knowlesi*.

## Ecological pressures driving introgression

In order to evaluate whether the cluster distribution or the introgression events are associated with ecological changes that might impact either macaque host or vector adaptations, we collated satellite-based surrounding forest fragmentation data and mosquito vector habitat suitability for 37 *P. knowlesi* samples in Sabah where village locations could be obtained (Figs A and B in S1 Text). These samples included 29 (78.4%) with two or more windows where introgression was observed, and 13 (35.1%) with high introgression (>5 windows).

Firstly, we performed univariate regression analyses of the *P. knowlesi* genomic clusters against proportional forest cover, intact forest perimeter-area ratio and *Anopheles* Leucosphyrus Complex mosquito vector habitat suitability metrics, with no statistically significant associations. Secondly, univariate regression analyses (optimal model as determined by AIC comparisons) suggested a limited relationship between two introgression windows, 859 (chromosome 08: 1280000 – 1289999) and 1236 (chromosome 10: 920000 – 929999) and the intact

forest perimeter-area ratio and mosquito vector habitat suitability, respectively (Tables E and F in S1 Text). The introgression window 859, which contains no identifiable genes on *PlasmoDB* and was identified in three Sabah and ten Sarawak isolates, was positively associated with intact forest perimeter-area ratio ($\chi^2 = 6$, df = 1, p = 0.02, $r^2 = 0.69$). The introgression window 1236, which was identified in eight Sabah and two Sarawak isolates, was negatively associated with the predicted mosquito vector habitat suitability ($\chi^2 = 8.17$, df = 1, p < 0.01, $r^2 = 0.71$). Putative genes overlapping this region include two encoding for unknown proteins, one encoding for ras-related protein Rab-1B and another for orotate phosphoribosyltransferase (Table 1).

## Investigation of antimalarial drug resistance candidates in *P. knowlesi* orthologues

The presence of antimalarial drug resistance determinants in *P. knowlesi* infections could be considered a surrogate marker of human-human transmission given the absence of drug pressure in the macaque hosts and fitness costs that are often associated with resistance-conferring alleles [59]. We therefore investigated the prevalence of non-synonymous variants in *P. knowlesi* orthologues of genes that have previously been associated with *P. falciparum* and *P. vivax* resistance to antimalarial drugs [48,49]. Within low complexity infections (n=152), six non-synonymous variants were detected within the *P. knowlesi* orthologue of *pvdhps* (PKA1H_140035100) (Table 2), which may be linked to sulphadoxine resistance, although very few studies associate genotype and phenotype [49]. The most common variants occurred at codon Y308H (58.4%) and K66E (11.9%). A G422S variant was present in 4.3% of samples overall, although was found exclusively in 25% of isolates from Peninsular Malaysia. Similar to previous work [59], 13 non-synonymous mutations were also detected within the *P. knowlesi* orthologue to *pvdhfr* (PKA1H_050015200) (Table 2). This includes 17 samples with greater than one mutation, and two samples with three mutations. Dihydrofolate-reductase mutations, associated with resistance to pyrimethamine, arise readily in both *P. falciparum* [60] and *P. vivax* [61,62]. The most common mutations were at codon N272S (97.9%) and E262D (22.7%), with N272S occurring 3 amino acid positions from the *P. vivax* orthologue. Six mutations were also observed exclusively in isolates from Peninsular Malaysia, with a mean frequency of 10.6%. Lastly, several non-synonymous mutations were also observed in the PKA1H_140054000 gene (Table 2), resulting in several amino acid changes near those observed in *P. vivax*, and associated with the multidrug resistance protein 1 (*pvmrp1*). This includes seven amino acid substitutions that occur <3 amino acids from non-synonymous

**Table 2. Summary of identified drug resistance orthologue mutations.**

| Gene | *Pv* gene ID | *Pk* gene ID | AA – *Pv* | AA – *Pk* | Anti-malarial |
|---|---|---|---|---|---|
| *pvdhps* | PVP01_1429500 | PKA1H_140035100 | C422W | G422S | Sulphadoxine |
| *pvdhfr* | PVP01_0526600 | PKA1H_050015200 | N273K | N272K[+++] | Pyrimethamine |
| | | | N273K | N272S[+++] | |
| *pvmrp1* | PVP01_0203000 | PKA1H_140054000 | I1620T | C1908S[+] | Multi-drug |
| | | | C1018Y | V1283I[+] | |
| | | | K542E | N620K[++] | |
| | | | R259T | G316D[+++] | |
| | | | R259T | G316S[+++] | |
| | | | T234M | M251K[++] | |
| | | | T234M | E250G[+++] | |

mutations present in *pvmrp1*, which has a potential yet unconfirmed role in primaquine failure in *P. vivax* liver-stage infection relapse. However, the biology and significance of these SNPs in these putative genes may not translate directly from *P. falciparum* and *P. vivax*.

Drug resistance orthologues from *P. vivax* (putative) and *P. falciparum* (none identified from *P. falciparum*) identified in this *P. knowlesi* dataset. [+]Designates the number of amino acid positions the identified *P. knowlesi* mutation is from the corresponding *P. vivax* orthologue mutation position (proximal mutations may retain the potential to cause similar downstream effects). AA: amino acid change; Pk: *P. knowlesi*. Pv: *P. vivax*.

## Discussion

This study expands current understanding of *P. knowlesi* population genetics by incorporating additional whole genomes from Sabah, a key transmission area in Malaysia. We identified distinct geographical subpopulations within *Mf*- and *Mn*-associated clusters, with evidence of introgression between these clusters potentially driving differentiation. Preliminary ecological-genomic analysis suggests possible associations between genomic patterns and environmental features affecting host or vector adaptation. Additionally, we detected non-synonymous mutations in antimalarial drug resistance-related orthologous genes arising *de novo* given the zoonotic transmission mode and lack of drug selection pressure [63,64].

Within-host *Plasmodium* genetic diversity reflects transmission intensity, with superinfections arising from multiple mosquito bites or co-transmission of related parasite strains in a single bite [11,65,66]. The prevalence of human polyclonal *P. knowlesi* infections were lower than in *P. falciparum* or *P. vivax* endemic regions [67,68], although are likely higher in natural macaque hosts [69]. The zoonotic nature of *P. knowlesi* complicates infection complexity, with multiple underlying parasite, host and epidemiological factors potentially influencing the establishment of successful erythrocytic replication of multiple inoculated *P. knowlesi* strains within humans. These factors include specific parasite proteins involved in human red blood cell invasion including PkDBPαII and PkNBPXa [70], transmission intensity and the relationship with parasite genetic diversity in macaque hosts, and the impact of land use change on mosquito distribution and host biting preferences [71,72]. Inter-infection diversity can also be exacerbated by recombination between genetically distinct parasites within the mosquito (coinfections) [11]. Being a natural reservoir for multiple zoonotic *Plasmodium* species, macaques have been shown to be co-infected with up to five simian *Plasmodium* species [69] and multiple *P. knowlesi* clones [73,74]. Although it cannot be confirmed with the methods used here, the isolates with distinct clones could represent superinfections. While there is no evidence of sustained human-to-human transmission to date [75], high-risk groups like forestry and plantation workers face greater exposure to infected vectors and reservoir hosts [9,10,76]. The five individuals harbouring multiple distinct clones could belong to these at-risk groups, warranting further research with integrated epidemiological and genomic datasets.

Both neighbour-joining and IBD-based cluster analyses identified the three major known *P. knowlesi* genomic clusters in Malaysia. The majority of new isolates from Sabah belong to the *Mf* cluster, aligning with reports of higher prevalence of *Mf*-derived infections and the restricted habitat of *M. nemestrina* in intact forests [22,44]. The low median IBD in the *Mf* cluster suggests high transmission intensity and genetic diversity, typical of endemic *Plasmodium* populations with minimal inbreeding. In contrast, the higher IBD values in the *Mn* cluster suggests greater parasite relatedness and possible inbreeding, however, as the median IBD reduces substantially when down sampling to the *Mn* cluster, these values may be skewed by population structure [14]. The broad ecosystem range and adaptability of *M. fascicularis* [44] may contribute to the *Mf* cluster's higher genetic diversity, which could hinder malaria control

efforts by enhancing the parasite's ability to adapt to environmental changes and broaden efficient zoonotic transmission scenarios.

Deforestation and agricultural expansion have altered macaque and *Anopheles* habitats, likely driving recent genetic exchanges in human infections [14,72,76,77]. Regression analyses revealed significant associations between two introgressed windows and both forest fragmentation (perimeter-area ratio) and habitat suitability of the *Anopheles* Leucosphyrus Complex mosquito vector, broadly supporting this hypothesis at a population-level. These genomic regions contain putative genes critical for parasite survival and transmission, such as the microneme-associated antigen, which facilitates erythrocyte entry [78], and the oocyst-expressed *cap380* gene, essential for vector-stage transmission, previously identified in Betong, Sarawak [14,55]. Apicortin protein, vital for cytoskeletal stability, replication, and host erythrocyte invasion in *P. falciparum* and *P. vivax*, was also identified [79,80]. The presence of several human- and vector-related genes in introgressed windows suggests strong selective pressure from both hosts. However, while these genes are well-characterized in other *Plasmodium* species, their functions may not directly translate to *P. knowlesi*.

Introgression events occurring across large geographic distances suggest independent occurrences driven by similar environmental drivers, like deforestation and shifting vector populations. This is supported by the non-overlapping introgressed windows among isolates from different geographic regions (Fig I in S1 Text). However, this integrated genomic and spatial analysis is limited by the small subset of isolates and landscape metrics used [81], as well as the lack of temporal alignment between *P. knowlesi* isolate collection and environmental data, especially when one considers ongoing deforestation, reforestation and land use change in Sabah. Future work should involve larger sample sizes and a systematic approach to landscape classification, including accounting for temporal land-use changes.

Sampling of *P. knowlesi* infections in Malaysian Borneo occurred across two large geographical areas. While macaque host infection prevalence and transmission intensity at a troop level is the likely primary driver of *P. knowlesi* population structure, environmental factors likely influence structure across heterogenous landscapes. To assess geographical impacts, we analysed the *Mf* and *Mn* clusters separately, comparing Sabah and Sarawak subpopulations. As expected, samples collected in closer proximity showed higher relatedness, albeit less so than across the three major genomic clusters. Notably, within *Mf* two Sabah *P. knowlesi* isolates with a high degree of introgression clustered with Sarawak samples. One individual was from a village in Kudat but has a history of recent travel to Hutan Long Pasia in Sipitang district for work, which is located close to the Sarawak border. However, the other individual has no history of recent travel, suggesting this could be the result of independent introgression events arising across regions or the small possibility of an onwards human transmission event.

The stronger relatedness between geographically proximal samples suggests ecological pressures, alongside macaque hosts, influence *P. knowlesi* genomes. $F_{ST}$ analysis of the *Mf* cluster identified several vector-related genes, also found in introgression analysis, including the oocyst-expressed *cap380 gene* on chromosome 08. This finding may suggest regional differences in the mosquito vector species within the *Anopheles Leucosphyrus* Complex [82] contributes to *P. knowlesi* subpopulation variation. In Sabah, human land use change has altered vector behavior, breeding sites, and biting preferences in the primary vector *A. balabacensis* [83]. Future studies may benefit from using cluster-specific reference genomes for $F_{ST}$ analysis [84], particularly for the *Mn* cluster.

As *P. knowlesi* transmission appears exclusively zoonotic [2], and therefore without drug-pressure, resistance mutations are unlikely to arise. The *dhfr* and *dhps* mutations may be associated with resistance to pyrimethamine and sulphadoxine, previously used to treat *P. falciparum* in Malaysia [85]. However, artemisinin-combination therapy is now the

recommended treatment for uncomplicated malaria in Malaysia, including *P. knowlesi*, eliminating the potential for ongoing sulphadoxine-pyrimethamine selection pressure [86,87,88]. We have also previously showed that *dhfr* mutations in *P. knowlesi* are unlikely to be due to sulphadoxine-pyrimethamine selection pressure due to not occurring in the drug binding domain [59], and no *dhps* mutations associated with resistance have been identified [17]. The absence of proven natural human-to-human transmission and sulphadoxine-pyrimethamine use for *P. knowlesi* infections, suggests that these mutations likely reflect the polymorphic nature of these genes rather than drug selection pressure. The highly prevalent N272S mutation in *dhfr* appears fixed in the population, with the reference allele (A) found only in older lab-adapted lines, originally collected in the 1960's, and the A1.H.1 strain, while the alternate allele (G) dominates recent populations and the PKNH reference genome [89,90]. Lastly, although *mrp1* variants have been reported [16], the functional impact of the SNPs observed here remains unclear, as the biology may not directly translate from *P. falciparum* or *P. vivax*.

The addition of 52 high-quality *P. knowlesi* genomes from Sabah, Malaysia enhances our understanding of this unique parasite's evolving genomic landscape. We identify polyclonal infections and describe novel regional *P. knowlesi* within-cluster subpopulations, likely driven by introgression between the *Mf-* and *Mn-*associated clusters. These genomic introgression events in turn may reflect ecological influences from host or vector adaptations. Human encroachment on ecosystems through anthropogenic deforestation and agriculture appears to align with these genetic changes. Additionally, non-synonymous mutations were found in *dhps, dhfr* and *mrp1* putative drug-resistant genes. Insights from *P. falciparum* and *P. vivax* highlight the importance of expanding and adapting integrated genetic, epidemiological and environmental surveillance efforts to address the zoonotic context of *P. knowlesi* when developing future public health control strategies.

## Supporting information

**S1 Text. Masked genomic regions and supplementary outputs.**
(DOCX)

## Acknowledgements

We thank the study participants, and the research team at the Infectious Disease Society Kota Kinabalu Sabah including Sitti Saimah binti Sakam, Azielia Elastiqah binti Salamth and Mohd Rizan Osman. We thank the Director-General, Ministry of Health, Malaysia, for permission to publish this manuscript. We thank Dr Freya Shearer and Dr David Duncan from the University of Melbourne for their consultation on specific analyses.

## Author contributions

**Conceptualization:** Jacob A.F. Westaway, Ernest Diez Benavente, Sarah Auburn, Roberto Amato, Nicholas M. Anstey, Zbynek Bozdech, Matthew Field, Matthew J. Grigg.

**Data curation:** Jacob A.F. Westaway, Sourav Nayak, Danshy A. Alaza, Matthew J. Grigg.

**Formal analysis:** Ernest Diez Benavente, Sarah Auburn, Edwin Sutanto, Hidayat Trimarsanto, Matthew Field, Matthew J. Grigg.

**Funding acquisition:** Timothy William, Giri S Rajahram, Bridget E. Barber, Chris Drakeley, Nicholas M. Anstey, Zbynek Bozdech, Matthew Field, Matthew J. Grigg.

**Investigation:** Jacob A.F. Westaway, Michal Kucharski, Nicolas Aranciaga, Sourav Nayak, Kim A. Piera, Kamil Braima, Angelica F. Tan.

**Methodology:** Jacob A.F. Westaway, Sarah Auburn, Roberto Amato, Nicholas M. Anstey, Zbynek Bozdech, Matthew Field, Matthew J. Grigg.

**Project administration:** Jacob A.F. Westaway, Sarah Auburn, Nicholas M. Anstey, Zbynek Bozdech, Matthew Field, Matthew J. Grigg.

**Resources:** Timothy William, Giri S. Rajahram, Bridget E Barber, Jenarun Jelip, Nicholas M Anstey, Zbynek Bozdech, Matthew Field, Matthew J. Grigg.

**Software:** Jacob A.F. Westaway, Ernest Diez Benavente, Sarah Auburn, Edwin Sutanto, Hidayat Trimarsanto, Matthew Field.

**Supervision:** Ernest Diez Benavente, Sarah Auburn, Nicholas M. Anstey, Matthew Field, Matthew J. Grigg.

**Validation:** Jacob A.F. Westaway, Matthew Field.

**Visualization:** Jacob A.F. Westaway, Ernest Diez Benavente, Sarah Auburn, Edwin Sutanto, Hidayat Trimarsanto, Matthew Field, Matthew J. Grigg.

**Writing – original draft:** Jacob A.F. Westaway.

**Writing – review & editing:** Jacob A.F. Westaway, Ernest Diez Benavente, Sarah Auburn, Michal Kucharski, Nicolas Aranciaga, Kamil Braima, Chris Drakeley, Edwin Sutanto, Hidayat Trimarsanto, Jenarun Jelip, Nicholas M. Anstey, Zbynek Bozdech, Matthew Field, Matthew J. Grigg.

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
