## [Decision Letter · Decision Letter 0]

29 Aug 2024

Dear Dr Westaway,

Thank you very much for submitting your manuscript "Genomic epidemiology of Plasmodium knowlesi reveals putative genetic drivers of adaptation in Malaysia." for consideration at PLOS Neglected Tropical Diseases. As with all papers reviewed by the journal, your manuscript was reviewed by members of the editorial board and by several independent reviewers. In light of the reviews (below this email), we would like to invite the resubmission of a significantly-revised version that takes into account the reviewers' comments. 

We cannot make any decision about publication until we have seen the revised manuscript and your response to the reviewers' comments. Your revised manuscript is also likely to be sent to reviewers for further evaluation.

Sincerely,

Nadira D. Karunaweera

Academic Editor

Paul Brindley

Editor-in-Chief

Reviewer's Responses to Questions

**Key Review Criteria Required for Acceptance?**

**Methods**

-Are the objectives of the study clearly articulated with a clear testable hypothesis stated?

-Is the study design appropriate to address the stated objectives?

-Is the population clearly described and appropriate for the hypothesis being tested?

-Is the sample size sufficient to ensure adequate power to address the hypothesis being tested?

-Were correct statistical analysis used to support conclusions?

-Are there concerns about ethical or regulatory requirements being met?

Reviewer #1: please see 'general comments'

Reviewer #2: (No Response)

Reviewer #3: The study by Westaway et al. is well-structured, with appropriate methods that answer key biological questions. However, several methodological points need clarification to improve the manuscript.

1. Re: sample collection. The authors should indicate whether the samples included in the study were single-species Pk infections. Were they tested for Pf, Pv, etc.? If not, why not? If so, how were co-infected samples dealt with? Were they excluded for ease of analysis? Did the laboratory or bioinformatic methods selectively amplify Pk vs other Plasmodium DNA?

2. Re: variant detection pipeline. The authors indicate that they used a modified version of a previously described workflow, but do not comment on what those modifications were. It appears that they largely followed GATK Best Practices, with a few notable deviations that are not explained:

a. Why was GATK 3.8 used instead of GATK 4, which has been available since 2018?

b. How was the "conservative set of high-quality variants" generated? What metrics were used to designate them?

c. The base recalibration and indel realignment method is unlike any I have seen before and does not follow GATK's best practices guidelines, which recommend using hard filtering in cases where there is no available variant database. What was the justification for using the chosen method rather than simply looking at the distributions of quality metrics (e.g. FS, QD, MQ) in the entire variant callset and choosing conservative cutoffs?

d. The decision to use both bcftools mpileup and GATK HaplotypeCaller is not explained. Presumably, the authors determined that variants identified by both methods were higher quality, but this should be spelled out as it is not a standard method.

3. Re: population structure. 

a. The chosen thresholds for IBD plots are very confusing, both in terms of the written description and in terms of the plots, which do not match the description. My best interpretation of the description is that the authors started with 95%, then halved it to 42.5%, then halved it again, which would be 21.25%. However, the actual cutoffs used are different, with no explanation as to why. In addition, the choice to do this at all is confusing. Why not just use standard cutoffs, e.g. 95%, 90%, 50%, 25%, 10%, etc.? The authors may have an intuitive sense of what IBD of 11.8% means biologically, but I would suspect they are in the minority if so.

b. CV error values from ADMIXTURE should be given, if only in a supplementary graph. Specifically, how different are the values for 3 populations vs. 2 or 4? The results are extremely convincing that there are 3 populations, but including these values would add additional support.

4. Re: characterizing subpopulations. The authors indicate that they controlled for population structure by subsetting samples, but they do not explain how this subsetting occurred. What were the thresholds? What was the subsetting scheme (i.e. was it proportional? random? data-driven?)?

**Results**

-Does the analysis presented match the analysis plan?

-Are the results clearly and completely presented?

-Are the figures (Tables, Images) of sufficient quality for clarity?

Reviewer #1: please see 'general comments'

Reviewer #2: (No Response)

Reviewer #3: The results presented in the manuscript are very compelling and paint a clear picture of Pk genomic epidemiology in Sabah. Nonetheless, as with the methods, there are several points that are unclear and/or insufficiently explained. In addition, while the figures overall are clear and well-presented, there is room for improvement as detailed below.

1. Re: coverage.

a. 30.27% of reads mapped to the Pk reference genome? What were the rest? Human? Macaque? Other Plasmodium coinfections?

b. How evenly distributed was coverage across the chromosomes? How variable was coverage between samples? The overall average is given, but not a range.

2. Re: clonality.

a. Fws is a sufficient metric for distinguishing monoclonal from polyclonal infections, however it is fairly arbitrary and less informative than COI/MOI. Particularly given that Fws was calculated in moimix, why did the authors not actually calculate MOI? It would also be much clearer which samples should be excluded for the plot in Figure 1 with MOI estimates. The decision to use Fws rather than MOI needs to be clearly justified.

b. What was the cutoff for a sample to be considered a superinfection?

c. In Figure 1, the authors have filtered out samples with Fws < 0.85 but are then making a statistical comparison. How many samples were excluded? Could this exclusion affect the statistical analysis? Or was the statistical test run before excluding samples with lower Fws? Here again, MOI is more mathematically tractable than Fws as well as easier to interpret.

3. Re: clustering. 

a. The ADMIXTURE data is very convincing and compelling. A PCA would also be beneficial to demonstrate further support for the geographic population structure.

b. The color scheme in Figure 2A is difficult to distinguish. The green and blue colors are insufficiently divergent in the context of this NJ tree. I would suggest the authors explore other, more divergent palettes in RColorBrewer or viridis.

4. Re: IBD. See my comment in the methods review. These cutoffs are not intuitive or consistent with the scheme outlined in the methods.

5. Re: introgression. 

a. Figure 4C has the same problem as Figure 2A.

b. Are the proteins in Table 1 based on Pf orthologs (i.e. are they all putative)? I suspect that their biological significance is likely unclear based on my own experience with this type of analysis, but it wouldn't hurt to be a little more explicit about the weight of the evidence here.

<b>Conclusions

-Are the conclusions supported by the data presented?

-Are the limitations of analysis clearly described?

-Do the authors discuss how these data can be helpful to advance our understanding of the topic under study?

-Is public health relevance addressed?

Reviewer #1: please see 'general comments'

Reviewer #2: (No Response)

Reviewer #3: The conclusions overall are well supported, with clear limitations described. However, see my comment in the "Summary and General Comments" section about appropriately contextualizing the findings.

**Editorial and Data Presentation Modifications?**

Reviewer #1: please see 'general comments'

Reviewer #2: (No Response)

Reviewer #3: There are some missing words, typos, and missing italics throughout the manuscript.

**Summary and General Comments**

Reviewer #1: This manuscript describes the addition of 52 whole genome sequences of Plasmodium knowlesi from Sabah state to a panel of previously sequenced genomes, and the in-depth population genomics analyses of the combined dataset. This well written account adds further insights to the current understanding of the population structure of Pk in Malaysia, and, the authors claim, reveals ‘genetic drivers of adaptation’ in this parasite. 

The manuscript describes complex population genetic analyses, which appear to give some fairly simple results; the previously described population structure of Pk in Malaysia holds up, and there are some mutations in drug resistance genes that were probably not selected by drug pressure. There is some evidence for introgression between the three populations, especially between the two populations previously characterised as representing distinct pools of parasites infecting Macaca nemestrina and Macaca fascicularis. 

I have one comment/question regarding the authors assumption that an infection with two very different clones must be the result of super-infection rather than co-infection. What piqued my interest in this, was the observation that all their supposed ‘superinfections’ were composed of two distinct clones. Please forgive my naivety here, but I wonder if the authors might clarify how they are able to differentiate between a parasite isolate that contains two distinct and unrecombined clones (a mixture) and one that contains the products of a genetic cross between the two same clones. To simplify this thought, if a monkey was infected with two distinct contemporaneously gametocyte producing strains of Pk, and was bitten by a mosquito, these would either undergo recombination in the mosquito resulting in 50% recombinant progeny and 50% ‘selfed’ parental strains, or they would not recombine, and the resulting sporozoites would be 50% of each parental strain. The latter scenario is unlikely, but could happen due to mating incompatibility or something. However, when you sequence the resulting blood stage infection, you would measure the same thing – 50% alleles of parent A, and 50% alleles of parent B. How is it possible to differentiate between them in this case? What am I missing here? I think what I’m struggling (badly) to convey, is that just because there are two seemingly unrelated alleles at each locus in a 2-strain infection, it doesn’t follow that these must have come from a superinfection. A co-infection is just as likely, isn’t it? Please bear in mind I’m coming at this from experience of making genetic crosses between two different strains of parasite in the lab, and finding out if we’ve got a ‘cross’ or a ‘mixture’ is relatively taxing. In both cases, of course, it was not a superinfection, but a coinfection that produced the mixed allele infection. 

So, how confident are the authors that the 2-clone human infections are superinfections, and not just the result of a person being bitten by a mosquito that previously bit a monkey with a 2-clone Pk infection (that very well could have been the result of a superinfection, as I bet these are extremely common in monkeys). 

I raise this point because it is interesting to me as a geneticist and an epidemiologist, but also because it impacts on one of the main points of discussion for the paper. Please clarify. 

Another, niggly (sorry) point. I think this paper expands our knowledge of what’s going on with Pk in Borneo, and commend the authors on carrying out some very clever work. But, to claim that the work is (or similar work could be) “invaluable in guiding future public health surveillance and control strategies” is reaching a bit, isn’t it? If such a claim is to be made that the results of this work can guide future control strategies, then this statement should be backed up with some real, solid examples of just how this will work. Which data/findings here will help control strategies? How will they do so? What differences in control strategies should be made after these results were generated compared to before? 

Further (extremely) minor points:

Abstract 

Line 66 – please expand “IBD” here

Author Summary

Line 84- I think it is very much the highest the highest burden in the country, and, indeed, the world. 

Line 85. I think the word “thus” is not needed here, as it doesn’t really follow. 

Introduction

Line 132. What gap? Suggest changing ‘this’ to ‘the’

Line 155: Perhaps mention the PCR target genes here to save a click for the reader?

Line 230 Any chance of a reference for Moran’s I?

Line 259: Same for Akaike Information Criterion

I appreciate these terms may be common and immediately understandable to modern population geneticists, but please be kind to the potentially more general readership of PLoS NTDs. In fact, the methodology in its entirety would benefit for some simple explanations of how the various methods actually work, and the concepts they’re based on, in layman’s terms. A hard task I know, but worth the effort!

Line 291: Could the dates ranges of collections be given here? Were samples collected contemporaneously for all populations? If not, is this a potential problem?

Line 301: What does ‘exclusion for minor allele frequency mean’?

Discussion:

Mainly, my concerns are with the assertion of superinfections, which I may be horribly wrong about.

Line 613 Why do you think these have ‘emerged’ (presumably you mean recently), and not been there for a long time?

The discussion is very long – I usually applaud this, but in this case it might be better to streamline it to get to the salient points and hammer home the main messages. It repeats a lot of what was already said in the results section. I’d like to see this section focus more on what the consequences of these findings are, practically, rather than rehashing the results. 

Reviewer #2: Jacob Westaway and colleagues present a very comprehensive analysis of P. knowlesi whole genome sequencing data, combining 52 new genomes and 100 published genomes from different parts of Malaysia. The manuscript is well written and relatively long, which is mostly justified due to the detailed analysis (though some suggestions to shorten and clarify the discussion are given). In some section, there appears to be a bit of a risk of overinterpretation of the data, details below. In some other sections, some analysis (or their descriptions) would benefit from further clarification.

Major comments

Lines 320 to 327, and Figure 1B: The logic of the text and figure is difficult to follow. Unless I misread Figure 1B, the plots show the relative proportions of clones in multiclonal infections. The plot at the bottom left represents a multiclonal infection with an approximately 90%:10% split, while the plot at center right represents an approximately 50:50 split. These plots do not say anything about the relatedness between the clones. This is only captured in the Fws data, which is a very small part of Figure 1B. 

(I am not sure whether a correlation between Fws and NRAF is expected or typically expected – if there is, please apologize my ignorance).

For Figure 1A, the text states that is shows different relatedness levels, but the figure only ranges from 0.87 or so to 1.0. I am confused as why the figure does not capture the full range of Fws values observed.

Lines 387-88: The following sentence is not correct: “At IBD thresholds under ~30%, three relatively distinct, large infection networks were observed”

At 11.8%, there are two networks, with the remaining genomes not being connected. It is not correct to define them as a network.

Lines 290-92: Am I supposed to see that in Figure 3? I am not sure whether the lines are invisible, or whether it is not displayed. (Edit: The figure legend answers this question (not shown as <11.8%), but still the main text could be clarified).

Also, it seems that at such a low threshold (3%), genomes are connected almost always. Is this truly a result, or simply an artifact? Could you run a statistical test on that? It seems unusual to comment on anything being related at the 3% threshold.

Lines 498-509 (and lines 695-690): I have a number of questions about this analysis. First, the hypothesis that introgressions are directly linked to vector habitats is far-fetched. Introgressions likely happened many transmission cycles before the sample was collected for sequencing, it is unknown whether at this point the parasite was transmitted in a similar habitat. Second, parasites collected closely together are expected to be closely related and likely carry the same introgressions, but are also more likely to share habitat characteristics. It appears to me that this confounder was not accounted for. Chi square tests require independence between observations. This was likely not the case here, resulting in invalid results. In my opinion, this analysis is error prone and adds little to the manuscript, and thus could be deleted.

Lines 515-543 and Figure 5: This section seems to be overlapping Figure 2B and C and associated texts. To improve clarity of the manuscript, I recommend to combine these sections and discuss the distribution and structure of Mf and Mn in Sabah and Sarawak only once.

Line 704: “…suggests introgression events could be arising independently”

In that case, would they have slightly different break points? If so, could you check whether this hypothesis is true?

Minor comments

Lines 60 and 85: Given Malaysia has among the highest Pk transmission in the world, would it be more adequate to say that in Sabah it is among the highest in the world?

Line 102: It seems a word is missing here. Maybe “In contrast to the control of other species…”

Line 105: “Plasmodium species” used above (not “Spp”), recommended to be consistent.

Line 135: Please add that out of the 108 genomes, 100 were included in the analysis.

Line 309: The title “High prevalence of polyclonal P. knowlesi infections in Malaysian Borneo, but infrequent superinfection” seems not to represent the data. First, I would question whether 10-17% is high. Second, the frequency of polyclonal infections is virtually identical in peninsular Malaysia and in Borneo, thus way stress out Malaysian Borneo?

Following this, lines 316-318 are confusing: “The highest proportion of polyclonal infections was observed in Sarawak (17.6%, n=13/74), followed by Sabah (10.6%, n=10/94), and Peninsular Malaysia (12.1%, n=4/33).” 12.1% is more than 10.6%, thus Sabbah does not follow Sarawak.

Lines 352-53: There is no possible way to split 16 genomes to come to come to 86.6% and 13.4%. Please check these numbers. 

I recommend making figure 2B larger. It is hard to see the dots on the map, and the legend next to it is also hard to read.

Lines 393-95: Please use either percentages or decimals for IBD, but not a mix.

Lines 465-85: It would be interesting to know whether the number of genes is higher than expected based on a random distribution. Given a large number of genes is involved in host interactions (in particular if considering both the human/monkey and mosquito hosts), the genes found might just be a random selection among those.

Line 659: “It is possible that the five individuals who were unfortunate enough to be inoculated multiple times belong to a high-risk demographic.”

It seems a bit unscientific to call this unfortunate, in particular as it is not clear whether high MOI results in more sever disease.

Lines 690-91: A part of the sentence seems missing.

Lines 696-97: This sentence states that the cap380 gene has only been identified in Betong. I doubt that’s what the authors want to say.

Line 724: I believe “genetic transfer” here means introgression(?) It would be clearer to use this term.

The discussion sometimes is a bit lengthy and lacking clarity. Some examples include:

Lines 656-58: This sentence is essentially a repetition/rewording of the preceding sentence (lines 654-55).

Lines 716-20: This repeats methods explained in detail before.

Lines 720-22: “As expected, within both the Mf and Mn clusters there is a higher degree of relatedness between samples collected in closer proximity, albeit less so when

compared to the three major genomic clusters.” I do not understand the part following “albeit…”

Lines 750-52: “The variant causing the highly prevalent N272S mutation within the dhfr gene, is only present in isolates from lab-adapted lines.” I am confused by this sentence. Is N272S highly prevalent or only present in lab-adapted lines? That seems to be mutually exclusive. 

Lines 755-57: I do not understand this sentence. SP will not treat an infection (it treats the patient), but kill the parasite. If it does so successfully, as stated here, there will be no surviving parasites carrying any mutations. 

Lines 771-785 are very repetitive. A single concluding sentence would be sufficient.

Typos

Line 84: P. knowlesi not in italics

Lines 364-65: Word missing: evidence of shared ancestry

Reviewer #3: Westaway and colleagues present a very interesting and convincing analysis of the genomics of an emerging and neglected malaria pathogen. This paper merits publication, and is a valuable contribution to the literature.

Strengths: Appropriate analyses are used, compelling and clear results are presented, and the data is well situated with regard to Pk biology. The writing is clear, and the figures are well-organized.

Weaknesses: Overall this is a strong paper. The weaknesses are largely related to unexplained methodological choices, as outlined in specific sections.

Major point:

The authors consistently refer to evidence from other Plasmodium species as they attempt to interpret biological significance in P. knowlesi. As a researcher used to working on non-model organisms, I strongly empathize. However, this evidence is not put into sufficient context. Plasmodium knowlesi is a distinct organism, which is not particularly closely related to any of the other Plasmodia from which evidence is derived. It should be made much more explicit throughout the results and discussion that results from falciparum or berghei in particular are not easily translated into biological significance in knowlesi. It is good and appropriate to mention this evidence from other species as the authors make their interpretation, but they should consistently acknowledge that the biological significance in Pk is unknown and cannot be assumed to be identical to that in culturable species with different natural hosts.

In addition, re: drug resistance in particular, the putatively resistant alleles in Pv need to be put into appropriate context, as being either purely association-based or based on orthologs in Pf. Therefore, since we cannot even assume that they play a biologically significant role in resistance in Pv, the connection to biological significance in Pk is even more tenuous. The authors are appropriately cautious in interpretation, but should soften their language even further, since there is no compelling empirical evidence for a role in Pk drug resistance.

Minor point:

I attempted to locate the uploaded sequence data on SRA, but was unable to find any record of it. I assume there was a typo or other minor mistake, but this needs to be rectified before publication.

Note: I completed this review in collaboration with a junior colleague to support their professional development.

PLOS authors have the option to publish the peer review history of their article (what does this mean? ). If published, this will include your full peer review and any attached files.

**Do you want your identity to be public for this peer review?** For information about this choice, including consent withdrawal, please see our Privacy Policy .

Reviewer #1: Yes: Richard Culleton

Reviewer #2: Yes: Cristian Koepfli

Reviewer #3: No
---

## [Decision Letter · Decision Letter 1]

14 Dec 2024

PNTD-D-24-00619R1Genomic epidemiology of Plasmodium knowlesi reveals putative genetic drivers of adaptation in Malaysia.PLOS Neglected Tropical Diseases Dear Dr. Westaway, Thank you for submitting your manuscript to PLOS Neglected Tropical Diseases. After careful consideration, we feel that it has merit but does not fully meet PLOS Neglected Tropical Diseases's publication criteria as it currently stands. Therefore, we invite you to submit a revised version of the manuscript that addresses the points raised during the review process. Please submit your revised manuscript within 30 days Jan 13 2025 11:59PM. If you will need more time than this to complete your revisions, please reply to this message or contact the journal office at plosntds@plos.org. Please include the following items when submitting your revised manuscript:* A rebuttal letter that responds to each point raised by the editor and reviewer(s). You should upload this letter as a separate file labeled 'Response to Reviewers '. This file does not need to include responses to any formatting updates and technical items listed in the 'Journal Requirements' section below.* A marked-up copy of your manuscript that highlights changes made to the original version. You should upload this as a separate file labeled 'Revised Manuscript with Track Changes '.* An unmarked version of your revised paper without tracked changes. You should upload this as a separate file labeled 'Manuscript '. If you would like to make changes to your financial disclosure, competing interests statement, or data availability statement, please make these updates within the submission form at the time of resubmission. Guidelines for resubmitting your figure files are available below the reviewer comments at the end of this letter. We look forward to receiving your revised manuscript. Kind regards, Nadira D. Karunaweera, MBBS, PhDAcademic EditorPLOS Neglected Tropical Diseases Paul J Brindley PhDEditor-in-ChiefPLOS Neglected Tropical Diseases

Shaden Kamhawi

co-Editor-in-Chief

Paul Brindley

co-Editor-in-Chief

 **Additional Editor Comments (if provided):** Please consider the additional points raised by reviewers  2 and 3, and revise the manuscript in response and/or provide a rebuttal.   **Journal Requirements:****Reviewers' comments:** Reviewer's Responses to Questions

**Key Review Criteria Required for Acceptance?**

**Methods**

-Are the objectives of the study clearly articulated with a clear testable hypothesis stated?

-Is the study design appropriate to address the stated objectives?

-Is the population clearly described and appropriate for the hypothesis being tested?

-Is the sample size sufficient to ensure adequate power to address the hypothesis being tested?

-Were correct statistical analysis used to support conclusions?

-Are there concerns about ethical or regulatory requirements being met?

Reviewer #1: (No Response)

Reviewer #2: (No Response)

Reviewer #3: (No Response)

**Results**

-Does the analysis presented match the analysis plan?

-Are the results clearly and completely presented?

-Are the figures (Tables, Images) of sufficient quality for clarity?

Reviewer #1: (No Response)

Reviewer #2: The revised manuscript iz greatly improved. A few points remain that should be addressed to include clarify.

Please add line numbers to future versions of the manuscript.

The first paragraph of the results explains the composition of the 152 samples included in the analysis. The second paragraph then presents data based on 201 isolates. It is difficult to understand why additional isolates were included in this analysis.

Figure legend 1: While a Manhattan Plot is not strictly defined, and thus technically it might not be wrong, the plot in this figure is rather different to a typical Manhattan Plot e.g. used in a GWAS. I would recommend not to use the term here.

When reporting introgression windows, it is unclear whether the authors refer to the number of introgressions (ie, biological events), or whether to the number of 10kb windows. For example, the authors state that “For the Mf cluster, chromosomes 8 (n = 35) and 11 (n = 21) had the greatest number of candidate windows, with several windows on chromosome 8 overlapping the large peak observed in the Fst analysis (Figure 5).” Does this mean there were 35 windows on chromosome 8 that were not adjacent, pointing to 35 independent introgression events? Or, is there a single window of 350kb, thus spanning 35 of the 10kb windows defined by the authors? If the latter, this is simply a result of the window size selected by the authors and does not have any biological meaning; rather the number of biological introgressions should be reported. Similar unclarity appears in other sentences.

“For Betong samples 85% (n =12/14) had high levels of introgression, compared to 50% of Papar samples (n=1/2), with all but one Betong sample from the Mf cluster.”

Is it really worth adding a comparison if one site has only two samples? Naturally, no conclusions can be drawn from this statement.

From what year were satellite images to study habitat suitability? Given samples were collected from 2011-2016, and the fast presumably deforestation in this part of the world, it seems important to use images from the time of sample collection. Please add the date of the satellite images, and maybe a comment on whether deforestation might have changed since samples were collected.

Page 27: What does “including 20 at high levels” mean?

Page 27: There seems to be a word missing in “suggesting that introgression events may be a contributing factor to regions of IBD.” > “High IBD”?

Reviewer #3: (No Response)

**Conclusions**

-Are the conclusions supported by the data presented?

-Are the limitations of analysis clearly described?

-Do the authors discuss how these data can be helpful to advance our understanding of the topic under study?

-Is public health relevance addressed?

Reviewer #1: (No Response)

Reviewer #2: (No Response)

Reviewer #3: (No Response)

**Editorial and Data Presentation Modifications?**

Reviewer #1: (No Response)

Reviewer #2: (No Response)

Reviewer #3: (No Response)

**Summary and General Comments**

Reviewer #1: (No Response)

Reviewer #2: (No Response)

Reviewer #3: I appreciate the authors' thoughtful responses to my and the other reviewers' comments. The figures are also much improved.

Two very minor issues:

1. The way the authors refer to identity by descent and identity by state needs to be standardized. Throughout the manuscript, it is variously written in all caps, all lowercase, or all lowercase with hyphens (which I believe is the most correct, but the decision is yours). There are also multiple instances where descent is misspelled as either "descent" or "decent".

2. "join-genotyping" should presumably be "joint-genotyping"

One comment for future consideration:

Re: clonality, it is well known that many of the COI/MOI estimators spuriously inflate the number of clones present in a given sample, but this issue is not insurmountable. I am fine with your paper proceeding as-is with Fws since it's a minor finding, and might be useful for your future work:

I exclusively work on non-falciparum species (though not Pk), so I don't buy the idea that Pk is uniquely unsuited to these methods. Filtering by WSAF, missingness, and coverage (specifically with samples that are high outliers) can help alleviate the overinflation issue. I have also generated more reasonable estimates with coiaf than with The Real McCoil, but both are robust and I have used both for non-falciparum species.

PLOS authors have the option to publish the peer review history of their article (what does this mean? ). If published, this will include your full peer review and any attached files.

**Do you want your identity to be public for this peer review?** For information about this choice, including consent withdrawal, please see our Privacy Policy .

Reviewer #1: **Yes: ** Richard Culleton

Reviewer #2: No

Reviewer #3: No

---

## [Editor Report · Decision Letter 2]

3 Feb 2025

Dear Dr Westaway,

We are pleased to inform you that your manuscript 'Genomic epidemiology of Plasmodium knowlesi reveals putative genetic drivers of adaptation in Malaysia.' has been provisionally accepted for publication in PLOS Neglected Tropical Diseases.

Best regards,

Nadira D. Karunaweera, MBBS, PhD

Academic Editor

Paul Brindley, PhD

Editor-in-Chief

Shaden Kamhawi

co-Editor-in-Chief

Paul Brindley

co-Editor-in-Chief

---

## [Editor Report · Acceptance letter]

Dear Dr Westaway,

We are delighted to inform you that your manuscript, "Genomic epidemiology of Plasmodium knowlesi reveals putative genetic drivers of adaptation in Malaysia.," has been formally accepted for publication in PLOS Neglected Tropical Diseases.

Best regards,

Shaden Kamhawi

co-Editor-in-Chief

Paul Brindley

co-Editor-in-Chief
